# Fifty Years of Development of the Skull Vibration-Induced Nystagmus Test

**Solara Sinno** [1,2,*], **Sébastien Schmerber** [3,4], **Philippe Perrin** [1,2,5] **and Georges Dumas** [1,2,3]

1    EA 3450 DevAH, Development, Adaptation and Handicap, Faculty of Medicine, University of Lorraine, 54500 Vandoeuvre-lès-Nancy, France; philippe.perrin@univ-lorraine.fr (P.P.); georges.dumas10@outlook.fr (G.D.)
2    Laboratory for the Analysis of Posture, Equilibrium and Motor Function (LAPEM), University Hospital of Nancy, 54500 Vandoeuvre-lès-Nancy, France
3    Department of Oto-Rhino-Laryngology, Head and Neck Surgery, Grenoble Alpes University Hospital, 38000 Grenoble, France; sschmerber@chu-grenoble.fr
4    Brain Tec Lab UMR1205, University Grenoble-Alpes, CHU Michallon, 38000 Grenoble, France
5    Department of Pediatric Oto-Rhino-Laryngology, University Hospital of Nancy, 54500 Vandoeuvre-lès-Nancy, France
*    Correspondence: solara.sinno@univ-lorraine.fr

**Abstract:** This review enumerates most of the studies on the Skull Vibration-Induced Nystagmus Test (SVINT) in the past 50 years from different research groups around the world. It is an attempt to demonstrate the evolution of this test and its increased interest around the globe. It explores clinical studies and animal studies, both permitting a better understanding of the importance of SVINT and its pathophysiology.

**Keywords:** Skull Vibration Induced Nystagmus Test (SVINT)

## 1. Introduction

The Skull Vibration Induced Nystagmus Test (SVINT) has gained attention because of its simplicity and efficiency. It is currently used as a common and easy first-line or bedside examination test and has been included as part of routine vestibular clinical examinations since 1999.

It has been described as a "vestibular Weber test" [1,2] and is also known as the Dumas test. This author published the largest series of patients explored (18,500 patients) in his Ph.D. thesis and has worked on the validation of the test and location/frequency optimization of the stimulus. The reliability and reproducibility of the test have been demonstrated by Park et al. [3]. A strong correlation between SVINT and caloric tests exploring the horizontal canal response has been confirmed by Japanese group [4] and European group [5]

Vibration-induced nystagmus (VIN) may be elicited either by cranial stimulations (bone-conducted vibrations—BCV) or muscular cervical stimulations [6,7]. However, BCV using the vestibulo-ocular reflex (VOR) are more efficient than cervical stimulations when applied to the lower part of the neck using the cervical ocular reflex (COR) and so for clarity the label (or designation) of Skull Vibration Induced Nystagmus (SVIN) is currently preferred to VIN to name the test when cranial vibrations are performed.

Skull vibration-induced nystagmus (SVIN) is the result of noninvasive 100 Hz cranial vibrations [1], which stimulate both otoliths and semicircular canals (SCC). Otolaryngologists use it to screen for vestibular asymmetry and uncover vestibular dysfunction [5,8]. The first mention of the effects of vibrations applied to the skull is attributed to Von-Bekesy in 1935 [9]. This author suggested that vibration applied to the skull induced reflexes and motion illusions linked to stimulation of vestibular receptors. A VIN was incidentally described for the first time by Lücke in 1973 [10]. A more systematic clinical utility was

suggested in 1999 by Hamann [8] and Dumas [5,11]. It was the consensus meeting of the International Society of Otoneurology (SIO) in Briançon, France, in 2006, SVINT (in French: TVO—Test de vibration osseux), which officially presented this test as an independent new vestibular test, which needed to be validated. It was described as a test using BCV at 60 or 100 Hz applied to the skull (both mastoids and vertex) for a short period (5–10 s). The contents of this meeting were reported in the article of The Annales ORL 2007 [12]. Normative values, topographic, and frequency optimization of the stimulus were developed by the Grenoble group [5,13,14]. In patients with unilateral vestibular weakness, eye movement recordings revealed SVIN with a rapid phase generally beating away from the affected side. In total unilateral vestibular lesions (TUVL), it is the intact side that is stimulated [13]. A SVIN occurs when there is an asymmetry between peripheral vestibular receptors and the nystagmus is beating toward the side of higher excitability in partial unilateral vestibular loss (PUVL) [15]. The usefulness of the test has been confirmed by numerous authors for Vestibular neuritis (VN), Menière's disease (MD), Labyrinthine commotion, and follow-up after intratympanic gentamicin or vestibular neurectomy or in superior semicircular canal dehiscence (SSCD) and rare central neurological diseases, [3,4,6,8,15,16]. In patients with central lesions, a VIN is usually not observed, except in the case of a unilateral lesion in the brainstem [12,17] located on the VOR pathway [6]. No significant nystagmus is generally observed in healthy subjects. The test has been used in adults and children with hearing loss and vestibular dysfunction [18].

In this article, we will go through the 50 years of research that lead to the recognition of this test used as a high-frequency vestibular Weber test in clinical practice and which has been validated in an important series of patients.

## 2. Early Years

### 2.1. Clinical Findings in Humans

In 1973, Lücke incidentally described that vibratory stimulation of the craniofacial bones induced a nystagmus caused by 100 Hz vibrations in patients with unilateral vestibular lesions (UVL) [10]. Lackner and Graybiel (1974) reported that vibrations (60 to 120 Hz) applied on different points of the skull by a vibrator held by the examiner, or the examined subject was likely generating illusions of visual or postural movement and sometimes nystagmus in normal subjects [19]. They signaled that frequencies at 40 Hz or 280 Hz were not efficient. These authors already hypothesized stimulation by vibrations of the lateral and vertical SCCs. Kobayashi et al. in 1988 observed with Frenzel's goggles in normal/healthy subjects used as controls, vs. two patients with a UVL, a nystagmus induced by a 125 Hz vibration applied to the neck. In these two Total UVL (TUVL) patients (vestibular labyrinthectomy), the responses were significantly higher than in controls. They remarked that Vibration stimulation of the neck can modify the slow phase velocity of the caloric test (CaT) response in UVL [20]. So, they underlined the role of cervical afferents in vestibular compensation.

### 2.2. Animal Model

In 1977, Young et al. showed that a vibratory stimulation of 125 Hz to 350 Hz applied to the skull of anesthetized monkeys modified the action potentials of cells in the inner ear (although they tested a large frequency range between 50 and 4000 Hz) [21] and suggested that it was most probably due to the direct action and mobilization of inner ear fluids and to the endolymph wave on the hair bundle, rather than to the ampulla crest mobilization. They were the first to observe that in the squirrel monkey, vibrations applied to the frontal region elicited responses (changes in discharges and obtention of tuning curves) on isolated fibers exiting from SCC and otolith structures previously identified electrophysiologically after stimulation in various planes of space (the animals were maintained on rotating or tilting plates). They highlighted that neurons with irregular discharges were more sensitive than those with regular discharges. They also observed that the elective frequency of excitation

for the fibers originating from lateral SCC was at 250 Hz and for fibers originating from the saccule or utricle was at around 500 Hz [21].

### 3. Years 1990–2000

*Clinical Findings in Humans*

Table 1 summarizes the main findings from 1990 to 2000 with the major contribution of the French [5,22], German [8,23], and Japanese groups [7]. However, it is important to highlight the article by Dumas et al., 2000 which reported the optimization of the stimulus during SVINT and described the interest of the test in 31 TUVL, 16 PUVL (VN, preoperative VS), and 19 central brainstem lesions. These authors report a lesional-type VIN beating away from the lesion side encountered in all TUVL patients and correct the frequency of 50 Hz initially announced by Hamann and Michel for the ABC vibrator as being 100 Hz after the expertise of the 3SR (Soil, Solids, Structures) laboratory of the National School of Hydraulics and Mechanics of Grenoble (ENSHMG) (E. Ouedraogo). This experimentation was conducted using the CMT 100 daN sensor at the currently named ENSE3 physic laboratory in France [5]. A minimum vibration amplitude of 0.2 mm was suggested and a frequency of 100 Hz was reported as preferable for effective testing. Vibratory stimulation of amplitude <0.1 mm and frequency <20 Hz were reported as poorly efficient or ineffective. The test positivity criteria were specified. A skull vibration inducing a bilateral vestibular stimulation acting as a vestibular Weber Test was evoked, and the frequency, amplitude, and topographic optimization of the stimulus to obtain a VIN is described. These authors signaled the absence of modification of the VIN in a long-standing compensated TUVL. The SVINT was defined as a test to reveal and measure a vestibular asymmetry. Figure 1 shows an example of SVINT tracing, which corresponded to the first example of a SVIN 3D recording reported by the French group [presented in 1997 at the Societé internationale otoneurologie (SIO) in Liege (Belgium)]. Other 3D recordings with very similar tracing were observed and reported by Karlberg in a unilateral vestibular loss in 2003 with a scleral coil recording [24]. Yagi (1996) showed similar 3D results with cervical stimulus [7].

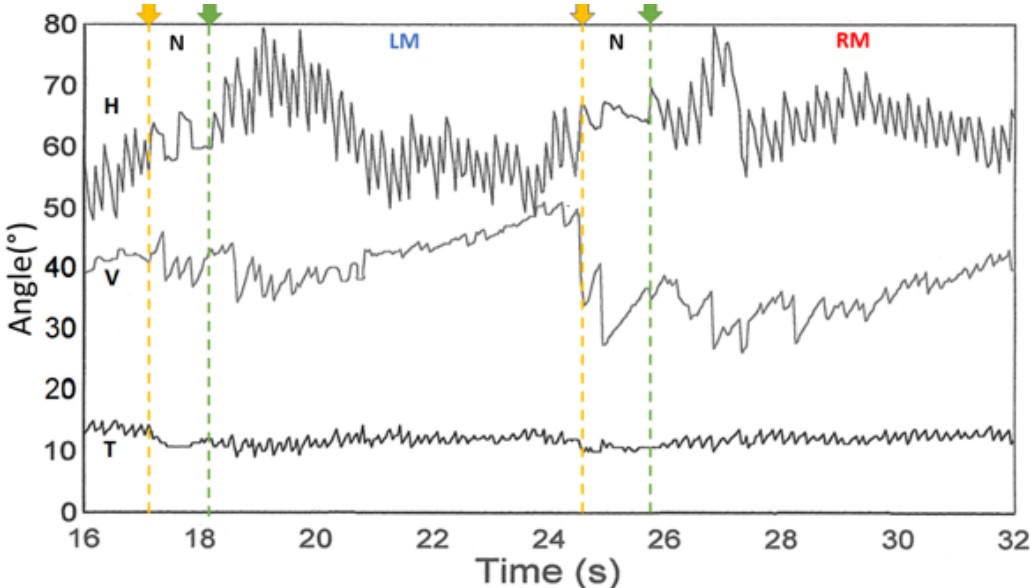

**Figure 1.** Three-dimensional recording of a positive test (BCV) (left severe unilateral vestibular lesion) from Dumas and Michel. Valeur sémeiologique du test de vibration osseux crânien. In: IPSEN, editor. XXXIème Symposium International d'Otoneurologie, Liège, Belgium (1997). The three components are observed H: Horizontal, V: Vertical, and T: Torsional. N: no stimulation; LM: Left mastoid stimulation; and RM: Right mastoid stimulation (VNG 3D recording device MFR-TOR3.4, Synapsys, Marseille, France).

Table 1. Summary of clinical findings between the years 1990 and 2000.

| French Group(s) | German Group(s) | Japanese Group(s) |
|---|---|---|
| • **Michel** in 1995 reported a series of 57 patients with peripheral vestibular involvement in whom stimulation was performed with the material of Hamann (ABC vibrator, Germany) signaled as delivering stimulation at 50 Hz and indicated its interest in cases of confirmed Menière's diseases (MD) or vestibular neuritis (VN) [22].<br><br>• **Dumas et al.** 1997 described nystagmus elicited by cervical muscle and bone (mastoids) stimulation vibrations in UVL cases and reported the finding at 100 Hz of a lesional type VIN beating toward the healthy side induced in a series of patients operated on for vestibular schwannomas (VS) by trans-labyrinthine route or in vestibular neurectomies for disabling MD (i.e., total vestibular loss: TUVL). Further details were provided on the positivity characteristics and the technical conditions of the VIN in two oral communications at the Strasbourg Congress in 1998 and report of the session of the French-speaking otoneurology society XXXIth Symposium of Liège Ed. Ipsen.<br><br>• **Dumas and Michel** 1999 [11] studied the semiological value of the cranial bone vibration test by 3D analysis of nystagmus. This work indicated the specificity and the sensitivity of the test in populations of total (n = 20 cases) and partial (n = 70 cases) lesions, its tolerance, and the presence of 3 components (horizontal, vertical, and torsional) of the VIN. Moreover, 100 normal subjects were explored, and specificity was 94%.<br><br>• **Dumas et al.** 2000 in an article on the optimization of the stimulus during SVINT described the interest of the test in 31 TUVL, 16 PUVL (VN, preoperative VS), and 19 central brainstem lesions [5]. In this article, the test positivity criteria were specified. | • **Strupp et al.** in 1998 described the changes of the horizontal eye position measured with VNG recordings (lateral ocular displacement of around 7° and subjective visual straight ahead (SVA) during vibration of neck muscles (stimulator applied 5 cm under the inion with a laterality of 2 cm on the right or an the left side (splenius and trapezius muscles), without systematic analysis of the nystagmus, in patients suffering from unilateral pathology (subacute period in 25 patients with unilateral vestibulopathy—VN explored at 15 days). They observed a significant ipsilateral displacement of the eye and SVA when ipsilateral muscles to the lesion where stimulated. They concluded that a unilateral increase in somatosensory weight substituted for missing vestibular input [23].<br><br>• **Hamann and Shuster** in 1999 [8], described a test capable of replacing the CaT, consisting of applying on mastoid processes vibrations at 60 Hz given as an optimal value in different vestibular pathologies (these authors previously signaled its interest in oral presentations in 1993 and 1995). This first clinical series of VIN provoked by the ABC vibrator (described as delivering optimal stimulation at 60 Hz) was reported in 60 patients with peripheral diseases (MD, VN, preoperative VS, BPPV, and otosclerosis) and in 47 central lesions. Seventy-five normal subjects were also explored. However, the optimal frequency was suggested to be at 60 Hz with the ABC vibratory. They also highlighted the VIN in MD might beat toward the affected side, but rarely and only in early irritative stages of MD patients. | • **Yagi and Ohyama**, in 1996, showed that vibratory stimulations of the cervical region in UVL patients induced a VIN beating towards the healthy side [7]. Moreover, a 3D analysis of the VIN also showed that the horizontal component was associated with an upward or downward vertical component after cervical stimulation (revealing a nystagmus of vestibular decompensation in subjects who were previously perfectly compensated). |

## 4. Years 2001–2010

### 4.1. Clinical Findings

A summary of the clinical findings of the diverse groups between the years 2001 and 2010 can be found in Table 2; highlighting the findings of the American group [25,26], Belgian group [27], French group, [26,28–30], Italian group [16,31–33], Spanish group [34]; Japanese group(s) [4], and Swedish group [35].

**Table 2.** Summary of clinical findings between the years 2000 and 2010.

| | |
|---|---|
| Australian Group(s) | **Karlberg et al.** (2003) used scleral coils to analyze 3D VIN at 92 Hz in patients with UVL (neurotomies, neuritis). They inferred that this nystagmus was secondary to an otolithic lesion or superior SCC implication [24]. The author suggested that the displacement of the SVH was secondary to a damage to the otolithic organ or superior SCC inducing an eye torsion. |
| American Group(s) | **White et al.** in 2007 described in eight patients with SSCD a down beating and torsional nystagmus induced by cranial vibration suggesting a direct stimulation of the dehiscent SCC [25]. |
| Belgian Group(s) | In 2008, **Boniver** hypothesized that SVINT was the result of altered proprioceptive inputs to neck muscles or direct stimulation of vestibular receptors in the intact labyrinth after unilateral vestibular deafferentation [27]. |
| French Group(s) | **Dumas et al.** published a report on the value of this test in clinical practice and recalled its fundamental bases [28]. Those clinical results were then presented at the Barany Society in Paris in 2004 and published in the special issue devoted to this society by the journal of vestibular research in 2004 [28]. **Dumas et al.** published in 2005 a series of partial vestibular lesions, signaling the interest and influence of stimulus frequency on the VIN and reporting for the first time in an article a SVIN in SSCD with a vertical but also a horizontal component [26]. **Michel et al.** reported the use of a 50 Hz (alleged frequency) vibrator in patients with confirmed MD. They assumed that VIN can be provoked by both the labyrinth and the neck muscle stimulations [29]. In 2004, **Ulmer et al.** provided additional information on the mechanisms involved by SVINT in case of UVL [30]. First, the vibrations of the skull selectively stimulated type I hair cells. Second, the direction of the beat of the nystagmus was toward the intact side/ear. They finally suggested that the vibrator excited both sides simultaneously, which suggested the primarily role of the intact side to provide the nystagmus, which reveals a vestibular asymmetry. |
| Italian Group(s) | **Nuti and Mandala** in 2005 studied the sensitivity and specificity of the mastoid vibration test in patients with VN using a handheld (Adele international Bologna Italy) battery powered device at 100 Hz. They concluded that the test had a sensitivity of 75% and specificity of 100%, and that the sensitivity of the test increased with increasing severity of the vestibular lesion and was well correlated with caloric paresis [16]. During the years 2008 and 2009, **Manzari** published articles in which he assessed different groups of patients and concluded that SVINT was useful in diagnosing SSCD, and in patients with otosclerosis (long stimulation of 40 s) with conductive hearing loss, it may be appropriate to evaluate the vestibular function. In SSCD cases, considering that the nystagmus was mainly rotatory or vertical, he hypothesized that it was in relation with the stimulation of the affected superior SCC and in otosclerosis, he hypothesized the horizontal nystagmus was linked to the ampullifugal/ampullipetal flow in lateral SCCs [31,32]. **Modugno et al.** in observed a positive SVIN in 44% of cases of the 86 schwannomas and a VIN beating ipsilaterally in 27% of cases [33]. |
| Japanese Group(s) | **Ohki** in 2003 [4] studied the VIN obtained after mastoid and frontal stimulations in patients with a UVL and compared it to the results of CaT and cervical vestibular evoked myogenic potential (cVEMP). This author demonstrated that the existence of a VIN with a dominant horizontal component, especially during mastoid stimulation beating toward the healthy side, was correlated with caloric unilateral weakness (when hypofunction was greater than 50%; a VIN was present in 90% of cases). He found no correlation with cVEMP. |
| Spanish Group(s) | In 2003, **Perez** published an article using a mini muscle massager and concluded that the value of the slow phase velocity (SPV) of vibration-induced nystagmus could be used to identify a sizeable proportion of patients with a vestibular disorder. In case of spontaneous nystagmus, the skull vibration enhanced the nystagmus SPV by 1.5 to twice the initial velocity [34]. |
| Swedish Group(s) | **Magnusson et al.** demonstrated that during bilateral vibration of neck muscles in normal subjects for posturographic recordings cervical muscle afferents played a dominant role over vestibular afferents, but they did not analyze concomitantly the SVIN [35]. |

*4.2. Animal Model*

Australian Group(s)

The Australian team showed that very high-frequency vibrations (500 Hz) induced the displacement of fluids, which were probably responsible of the deflection of cilia of the type I striolar receptors of the macula [36]. This irregular activated otolith afferent caused stimulus-induced eye movements in animals and humans (nystagmus). Curthoys hypothesized that vibrations, similar to sounds, caused pressure waves of fluid in the inner ear that mobilized the cilia and the hair-bundle at the apex of the hair cells of the vestibular receptors and induced the activation of type I receptors (at the striola level) and action potentials phase locking in irregularly discharging vestibular afferents, which synapsed to type I receptors [37,38].

Curthoys et al. defined the structures of the inner ear affected by vibration. In 2006, they studied the effect of vibrations driven by skull and bone conduction (B71) to determine the causative vestibular structures (SCCs and otoliths) in guinea pigs [37]. They studied the spontaneous discharge of simple primary vestibular neurons. In cases of angular acceleration, they categorized it as SCC neurons, and if it responded to sustained roll or pitch tilts then it was categorized as otolith neurons. Their experience showed that when the skull was vibrated (continuous pure tone at 200–1500 Hz and tone burst at 500 Hz), 4.7% of canal neurons, 14.1% of regular otolithic afferents, and 82.8% of irregular otolithic afferents responded. The majority of the afferent otoliths were in the superior division of the vestibular nerve and were probably of utricular origin. As a reminder, there are two types of cells in vestibular receptors: type II cells are phylogenetically older columnar cells present in amniotes (mammals, reptiles, and birds) and nonamniotes, while type cells I are vial-shaped and are found only in amniotes [37]. Types I have a greater number of calcium-activated potassium channels, which contributes to their greater sensitivity to high-frequency stimulation. In addition, type I hair cells provide inputs to the vestibular-irregular nerve afferent fibers, while type II hair cells are connected to regular afferents [37].

## 5. From 2011 until Now

*5.1. Clinical Findings*

5.1.1. American Group(s)

In 2015, White et al. published a case series of patients with enlarged vestibular aqueduct. They observed that after suboccipital vibrations, some patients had unidirectional horizontal nystagmus [39]. Zhang et al. explored a series of 812 patients, complaining of vertigo and dizziness, with SVINT, CaT, and ocular and cervical vestibular evoked myogenic potential. They confirmed that SVIN was a useful indicator of the asymmetry of vestibular function between the two ears when making judgments about SCC asymmetry but was less sensitive to show an asymmetry in otolith organ function [40]. Recently in 2021, Lin et al. showed that mastoid vibration while assessing posture, can be used to quantify the effect of deterioration by aging via posturographic recording [41].

5.1.2. French Group(s)

Dumas et al. in 2011 showed that in partial unilateral peripheral vestibular lesion the caloric test and SVINT correlated positively in 78% of the cases [15]. These same authors, in 2013, demonstrated that in chronic and compensated severe unilateral vestibular lesions (SUVL), SVIN beat toward the healthy side without any change when subjects had eyes closed on posturography [42]. Moreover in 2014, these authors showed that in otosclerosis cases, when patients were stimulated with short mastoid stimulations, 39% had a low amplitude positive SVIN, mostly beating toward the healthy side, rarely when the vertex was stimulated [14]. These authors proposed the clinical value of SVIN as a vestibular Weber Test [14]. In SSCD, this concept was emphasized in a further work published in 2017 in a review paper [6]. SVIN horizontal and torsional components are beating toward the side of the lesion in 95% of cases of unilateral SSCD was summarized in 2019 [1]. The vertical component was most often up beating (60%). The optimal frequency analysis

performed with Bruel and Kjaer 4810 Mini Shaker showed higher responses were observed at around 500 Hz. Responses were more frequently observed on the vertex than in patients with SUVL or TUVL (where they were mainly obtained on the mastoid process) at 100 Hz [1]. Sinno et al. highlighted the importance of the test as a screening tool in children with hearing loss (amplified with hearing aid or cochlear implant) [43]. In 2021, Dumas et al. found that SVINT results in a human model of horizontal SCC plugging used for disabling MD patient correlated with vestibular tests exploring horizontal canal function but not with cVEMP [44].

### 5.1.3. Chilean Group(s)

Waissbluth and Sepulveda also provided a systematic review published in 2021 and signaled the interest of a SVIN reproducible on both mastoid and with a SVIN-SPV >2°/s to confer a positivity criterion to the test [45].

### 5.1.4. Chinese Group(s)

Xie et al., in 112 patients with peripheral unilateral vestibular lesions, showed that a VIN was more frequently observed in case of SCC paresis with higher caloric deficit. They mentioned that SVIN horizontal component was beating toward the healthy side in 92% of the cases [46].

### 5.1.5. German Group (s)

Hamann proposed that the optimal frequency of stimulation with a customary device (Autronic, Hamburg, Germany) was 40 Hz [17]. This statement was different than all other reported publications [24–47].

### 5.1.6. Italian Group(s)

Teggi et al. in 2020 reported a SVIN in 58% of their 500 patients with confirmed MD most often beating away from the lesion side (98% of cases) [47]. They hypothesized that in patients reporting only photo-phonophobia during vertigo attacks and with a positive SVIN, the clinical manifestations may be predictive for evolution toward an MD, while migrainous headache and positive positional tests with negative SVIN were more frequently correlated to vestibular migraine [47,48].

### 5.1.7. Japanese Group(s)

Fujimoto et al., in 2021, studied the association between vestibular function and the findings of horizontal head-shaking nystagmus (HHSN) and SVINT tests. They concluded that HHSN had an association with LSCC dysfunction alone. SVINT had an association with dysfunction in all the SCCs and the utricle [49].

### 5.1.8. Portuguese Group(s)

In 2020, Matos et al. compared the results of VHIT and SVINT (100 Hz) at the time of the acute peripheral vestibular lesion and at the postacute phase in patients diagnosed with peripheral vestibular lesion (VN). They concluded that there is a clear contribution of the vertical SCC (confirmed by VHIT) in the presence of the vertical component of nystagmus in SVINT [50].

### 5.1.9. Spanish Group(s)

Vargas Gamarra in 2018 found that in cases of acute vestibular deficit and VN the response may diminish over time due to central compensation, but it rarely disappears (possibly due to the resolution of the vestibular asymmetries related to compensation) [51]. Matos and Perez studied in a population of acute vestibular lesions the horizontal and vertical component of SVIN and compared the result to those obtained with VHIT. They observed a global relation between all vertical SCCs results at the VHIT and the SVIN vertical component [50]. Martin-Sanz et al. in 2021, emphasized the efficiency of SVINT in

vs. cases and described a sensitivity and specificity at 81.8% and 73.9%, respectively [52]. Moreover, Batuecas-Caletrío et al. described a close relationship between the SVIN-SPV at 100 Hz and the difference of VHIT gains between the two ears [53]. Zamora et al. described the parameters of SVINT in normal subjects and proposed a discriminant SVIN-SPV value of 2.2°/s [54].

### 5.1.10. South Korea Group(s)

Koo et al. in 2011 analyzed the sensitivity of this test by comparing its results with spontaneous nystagmus and to other tests (head shaking test, spontaneous nystagmus test, and CaT) in the yaw axis in UVL patients. They noted that the VIN was as efficient as CaT to indicate lateralization (86%); and more efficient than the HSN. The nystagmus was beating toward the healthy side in 98% [55] In addition, Lee et al. showed that in VN, stimulation of the mastoid processes or sternocleidomastoid (SCM) muscles were equally effective for detecting vestibular asymmetry. The SVIN-SPV measuring degree of vestibular asymmetry was correlated with the volume of VS. when stimulation was applied to the very upper part of SCM muscle (2.5 cm under the mastoid process) [56]. However, possible radiation to the cranium and mastoid from this close cervical location remains possible to explain equivalent results observed with mastoid stimulation, as previously discussed by Dumas et al. [57].

### 5.2. Animal Models

Vulovic and Curthoys showed that bone conducted vibration activated the vestibulo-ocular reflex in a guinea pig [58]. Dlugaiczyk, in 2020, published an article emphasizing the fact that 500 Hz BCV is a largely selective otolithic stimulus, while 100 Hz BCV, used in SVINT, can activate both otolith and SCC afferents [59]. They also noted that the 100 Hz BCV did activate irregular SCC afferents in healthy animals with normally encased labyrinths, and they fire in a cycle-by-cycle basis so that 100Hz vibration causes a firing rate of 100 spikes/s, probably similar to that caused by a modest angular acceleration. This cycle-by-cycle activation of single canal afferent neurons to 100 Hz stimulation explains the abrupt start and stop of VIN and the lack of adaptation during the stimulus, and the lack of after-effects after the end of the stimulus.

### 6. Summary of the Agreement and International Consensus

- **Topographic stimulation:** International consensus relies on the mastoids position [6,45], with some interest in the vertex position (mainly French and German teams).
- **Frequency consensus:** 100 Hz widely used (majority of publications [3,4,7,10,16,24, 25,30,31,34,40,45–47,49,52,54,56]). The systematic study of frequency optimization, analyzing SVIN SPV in response to 10 Hz up to 800 Hz, performed by Dumas et al. with the Mini-shaker device of B&K has confirmed this optimal frequency empirically accepted by many other authors [1,6] (Figure 2)
- **Inner ear structure contribution to the constitution of the nystagmus (SVIN):** the SCC and particularly the horizontal SCC is the predominant structure affected by the 100 Hz vibration. The vertical SCC are responsible for the vertical and torsional component. [14,40,60]. As for the otoliths, the utricle may contribute in a small proportion, and the saccule is more controversial [24,40,44,61].
- **Test positivity criteria**: the nystagmus is reproducible, evoked during the stimulus/sustained response; disappears upon stimulation withdrawal; its slow phase velocity (SPV) >2–2.5 degrees/s, beating toward the same side/non-direction changing [1,3,45–47].

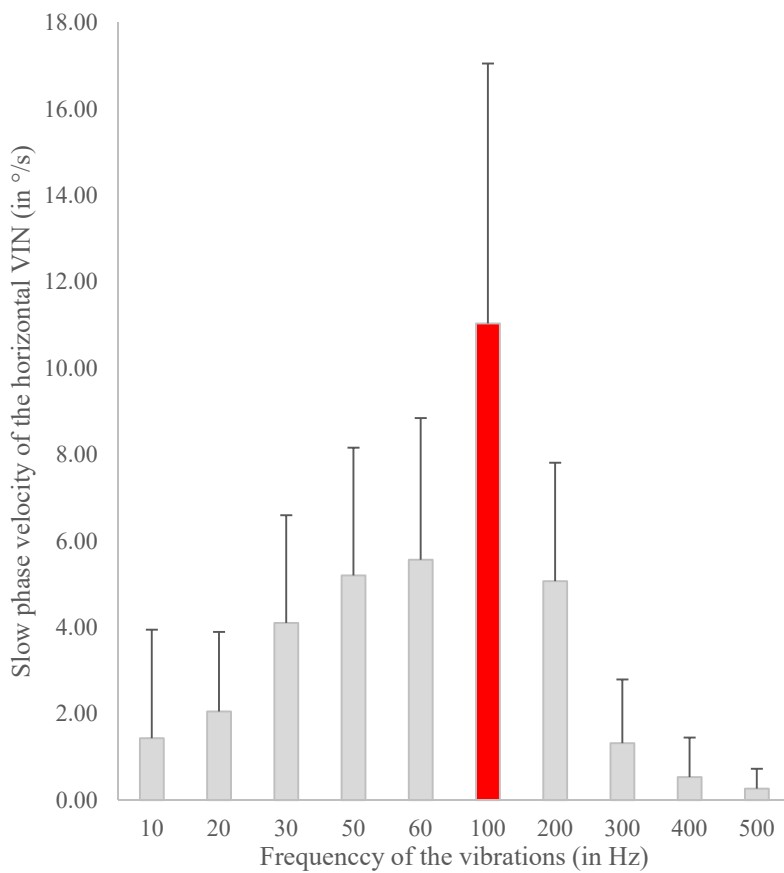

**Figure 2.** Frequency optimization (SVIN horizontal component) in 15 Total Unilateral Vestibular Loss patients (Translabyrinthine approach or Vestibular neurectomy). Stimulation with the Minishaker (Bruel Kjaer (B&K), Naerum, the Netherlands). Presented by Dumas et al. at the American Academy of Otolaryngology–Head and Neck Surgery (AAO-HNS) Congress, New Orleans, LA, USA, 2019 [62].

## 7. Conclusions

A VIN incidentally described from a clinical point of view by Lücke in 1973, has been evolved by different teams in the world. It may have a cervical (COR) or cranial (VOR) origin, however, for clinical practice, mastoid BCV were found more efficient to induce a stronger VIN response than cervical (lower part of the posterior neck) stimulation and cranial BCV stimulations are more clearly documented by physiology to allow a better clinical interpretation and understanding of the test. This justified the consensus name of Skull Vibration Induced Nystagmus Test (SVINT).

Its reproducibility and consistency were specified by Park. The correlation with horizontal SCCs and caloric test was highlighted by Ohki, Hamann, and Dumas. Its validation and stimulus frequency and location optimization as its criteria of positivity have been specified by the French team of Dumas et al. The SVINT interest from clinical practice has been illustrated by numerous and different teams in numerous peripheral pathologies as a bedside examination test. The fundamental bases and physiological background were given by different authors [21,63,64] who confirmed the implementation of type I inner ear hair cells, the direct stimulation of hair bundle at such high frequencies the concern of the transient system, and the stimulation of the VOR. This test initially not accepted and snubbed is now part of common bedside examination tests and globally used. It has become a reference for tests exploring very high frequencies and is now accepted as a screening high frequency vestibular Weber test to reveal a vestibular asymmetry.

**Author Contributions:** All authors have been part of the conceptualization and writing—original draft preparation, writing—review and editing. Supervision by G.D. and project administration S.S. (Solara Sinno). All authors have read and agreed to the published version of the manuscript.

**Funding:** This research received no external funding.

**Institutional Review Board Statement:** Not applicable.

**Informed Consent Statement:** Not applicable.

**Data Availability Statement:** Not applicable.

**Acknowledgments:** We thank Christol Fabre for figures.

**Conflicts of Interest:** The authors declare no conflict of interest.

**Abbreviations**

| | |
|---|---|
| **CaT** | Caloric Test |
| **MD** | Menière's disease |
| **UVL** | Unilateral vestibular lesion (PUVL = partial UVL, SUVL = severe UVL, TUVL = total UVL) |
| **SCC** | Semicircular canal |
| **SVIN** | Skull Vibration Induced Nystagmus |
| **SVINT** | Skull Vibration Induced Nystagmus Test |
| **VN** | Vestibular Neuritis |
| **VS** | Vestibular Schwannoma |
| **VIN** | Vibration Induced Nystagmus |
| **VOR** | Vestibulo Ocular Reflex |

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
