# Peer review of "Fifty Years of Development of the Skull Vibration-Induced Nystagmus Test"

_audiolres, doi:10.3390/audiolres12010002_

Round 1

Reviewer 1 Report

This paper adopts a very unusual approach to reviewing the literature on a clinical phenomenon -  by geographical location of the published papers rather than by the actual intellectual content of the research reported.  The result is a broad international view of the acceptability of SVIN across the world, which is of interest, however in my opinion this does need to be enhanced by providing also a summary of the agreement between different countries in the major results of this work.  

As the test developed there has been increasing awareness of the importance of factors such as vibration frequency a stimulus location.  I think a revision which takes greater account of the neural basis of SVIN would be useful.

Author Response

Dear Reviewers,

We thank you for your interest in our work and for your comments that greatly improved the manuscript. In this new version of the manuscript, the minor modifications requested by all 3 reviewers were all made. The highlighted points were taken into considerations, please find our point-by-point answers.

We hope the revised manuscript will better suit the journal and our reviewers.

With all our respect,

Comments from reviewer 1

This paper adopts a very unusual approach to reviewing the literature on a clinical phenomenon - by geographical location of the published papers rather than by the actual intellectual content of the research reported.  The result is a broad international view of the acceptability of SVIN across the world, which is of interest, however in my opinion this does need to be enhanced by providing also a summary of the agreement between different countries in the major results of this work.  

Answer: We have addressed the comment regarding the « summary of the agreement and the consensus » about the most important points, by including an additional paragraph at the discussion level.

As the test developed there has been increasing awareness of the importance of factors such as vibration frequency a stimulus location.  I think a revision which takes greater account of the neural basis of SVIN would be useful.

Answer: We have addressed this comment regarding the neural basis of SVINT, by developing this concept in the modified submitted version - section 5.2.

Reviewer 2 Report

This paper is pleasantly readable and shows a complete panorama of the evolution of thought regarding the vibratory test (SVINT). Only a typo on line 104 on the term Japanese.

Author Response

Dear Reviewers,

We thank you for your interest in our work and for your comments that greatly improved the manuscript. In this new version of the manuscript, the minor modifications requested by all 3 reviewers were all made. The highlighted points were taken into considerations, please find our point-by-point answers.

We hope the revised manuscript will better suit the journal and our reviewers.

With all our respect,

Comments from reviewer 2

This paper is pleasantly readable and shows a complete panorama of the evolution of thought regarding the vibratory test (SVINT). Only a typo on line 104 on the term Japanese.

Answer: we would like to thank the reviewer for his positive comment, and we have corrected the typo (term Japanese) in the new manuscript.

Reviewer 3 Report

The paper is a report of the history of SVINT. All important works on the topic have been considered and included.

I've only minor concerns on this paper

  • Probably the most important: review the  references, in some cases should be corrected including numbe of the volume and pages
  • I suggest to include in the paper before Introduction a list of abbreviation so to facilitate readers
  • In the introduction, when saying "In patients with unilateral..." including  a torsional component is often visible
  • Paragraph 5.2 might be expanded

Author Response

Dear Reviewers,

We thank you for your interest in our work and for your comments that greatly improved the manuscript. In this new version of the manuscript, the minor modifications requested by all 3 reviewers were all made. The highlighted points were taken into considerations, please find our point-by-point answers.

We hope the revised manuscript will better suit the journal and our reviewers.

With all our respect,

Comments from reviewer 3

The paper is a report of the history of SVINT. All important works on the topic have been considered and included. I've only minor concerns on this paper

  • Probably the most important: review the references, in some cases should be corrected including number of the volume and pages

Answer: The authors have reviewed the references and added the volume and pages in the new manuscript

  • I suggest to include in the paper before Introduction a list of abbreviation so to facilitate readers

we have addressed the comment regarding abbreviation and have added it at the beginning of the article.

  • In the introduction, when saying "In patients with unilateral..." including  a torsional component is often visible
  • Paragraph 5.2 might be expanded

We have addressed the comment by expanding the section 5.2.